# A Five Collagen-Related Gene Signature to Estimate the Prognosis and Immune Microenvironment in Clear Cell Renal Cell Cancer

**DOI:** 10.3390/vaccines9121510

**Published:** 2021-12-20

**Authors:** Xiaokai Shi, Xiao Zhou, Chuang Yue, Shenglin Gao, Zhiqin Sun, Chao Lu, Li Zuo

**Affiliations:** 1Department of Urology, The Affiliated Changzhou No. 2 People’s Hospital of Nanjing Medical University, Changzhou 213000, China; shixiaokai@njmu.edu.cn (X.S.); yuechuang9897@163.com (C.Y.); gsl_cmu@163.com (S.G.); deerchao@163.com (C.L.); 2Department of Oncology, The Affiliated Changzhou No. 2 People’s Hospital of Nanjing Medical University, Changzhou 213000, China; zhouxiao@stu.njmu.edu.cn; 3School of Clinical Medicine, The Affiliated Changzhou No. 2 People’s Hospital of Nanjing Medical University, Changzhou 213000, China; szq719819@163.com

**Keywords:** collagen, epithelial–mesenchymal transition (EMT), GSEA, immune microenvironment, prognosis

## Abstract

Collagen is the main component of the extracellular matrix (ECM) and might play an important role in tumor microenvironments. However, the relationship between collagen and clear cell renal cell cancer (ccRCC) is still not fully clarified. Hence, we aimed to establish a collagen-related signature to predict the prognosis and estimate the tumor immune microenvironment in ccRCC patients. Patients with a high risk score were often correlated with unfavorable overall survival (OS) and an immunosuppressive microenvironment. In addition, the collagen-related genetic signature was highly correlated with clinical pathological features and can be considered as an independent prognostic factor in ccRCC patients. Moreover, GSEA results show that patients with a high risk grade tend to be associated with epithelial–mesenchymal junctions (EMT) and immune responses. In this study, we developed a collagen-related gene signature, which might possess the potential to predict the prognosis and immune microenvironment of ccRCC patients and function as an independent prognostic factor in ccRCC.

## 1. Introduction

Renal cell carcinoma is the most common kidney malignant tumor, accounting for about 2% to 3% of adult malignant tumors. Among them, clear cell renal cell carcinoma is the most common type of renal cell carcinoma with its morbidity reaching up to 70–80% [1]. In recent years, the incidence and fatality rate of renal clear cell carcinoma have increased over the world [2]. Nowadays, due to the widespread use of imaging technology, the early diagnosis rate of renal clear cell carcinoma has increased significantly. However, about one-third of renal clear cell patients have been accompanied by distant metastasis at the time of initial diagnosis [3]. Currently, surgical resection is the main treatment for early localized clear cell renal carcinoma, but even with radical or partial nephrectomy, local or distant metastases still occur in 16% of patients [4]. For advanced metastatic clear cell renal cell carcinoma, because patients are not sensitive to radiotherapy and chemotherapy, the main treatment is targeted therapy and immunotherapy. However, 30% of patients with metastatic clear cell renal cell carcinoma have primary drug resistance toward molecularly targeted drugs and some patients will develop secondary drug resistance about 1 year after receiving treatment, which ultimately leads to a poor prognosis of patients [5]. Therefore, it is of great significance to explore the molecular mechanisms related to the progression of renal clear cell carcinoma for improving the diagnosis and prognosis of patients with renal clear cell carcinoma.

As is known to all, the seed–soil theory was proposed to elucidate the development process of tumors, which is a dynamic process of interaction between tumor cells and their microenvironment [6,7,8]. The tumor microenvironment is considered as a heterogeneous tissue including not only tumor cells but also immune cells, fibroblast, and endothelial cells. Moreover, the noncellular component extracellular matrix (ECM) also plays a critical role in this environment. In particular, the collagenous matrix is subjected to significant alterations in its composition and structure that create a permissive environment for tumor growth, invasion, and dissemination [9]. As the major component of the ECM, collagen functions as the attachment and scaffold for cell growth, which not only induces the proliferation, differentiation, and migration of epithelial cells but also plays an important role in maintaining intercellular adhesion, tissue integrity, and repairing as well as supporting organs [10]. Recently, increasing evidence showed that it plays a critical role in cancer progression and metastasis. For instance, collagen structural parameters (density, length, and width) and the presence of immature collagen molecules were reported to be associated with unfavorable outcomes of gastric cancer [11]. Furthermore, research demonstrated that excessive collagen deposition could reshape the tumor environment, thus defining cell fate and promoting tumor progression. Previously, the deposition of collagen was considered to be mainly released by cancer-associated fibroblasts (CAFs) [12]. However, increasing evidence has shown that collagen could also be produced by tumor cells in a specific manner. For example, a high expression level of collagen XIII was observed being produced in triple negative breast cancer cell lines [13]. These phenomena (collagen could be produced by cancer cells) were also observed in lung adenocarcinoma, esophageal squamous cell carcinoma cells, and so on [14,15]. Furthermore, evidence has demonstrated that the density and structural alteration of collagen I fiber could function as highways for cancer cells and macrophages for rapid travelling, thus resulting in cancer progression and metastasis [16,17,18,19].

In addition, accumulating evidence has shown that collagen expression could serve as prognostic indicators of cancer progression and metastasis. Overexpression of COL11A1 was reported to be associated with a relapse of non-small cell lung cancer [20]. In addition, upregulated COL6A1 in clear renal cell carcinoma and pancreatic cancer was reported to be a predictive prognostic biomarker [21,22]. Moreover, in head and neck squamous cell carcinoma, collagen was considered to be correlated with the alleviation of apoptotic responses to cisplatin via discoidin domain receptor 1 (DDR1) [23].

However, to date, the specific mechanism between ccRCC and collagen remains unclear. Hence, in this study, the collagen-related genes were comprehensively sorted out to establish a gene signature that might serve as a prognostic biomarker to assist the diagnosis of and improve the prediction of prognosis in clear cell renal cell cancer. This risk model might also be used to predict the immune microenvironment in ccRCC patients. These results will deepen our understanding between collagen and ccRCC, and the immune microenvironment.

## 2. Materials and Methods

### 2.1. Data Obtaining and Preprocessing

The transcriptome RNA sequencing data (FPKM value) and corresponding clinical and pathological information of 537 ccRCC patients and 72 normal tissue samples can be possessed from the TCGA (https://portal.gdc.cancer.gov/, accessed on 2 July 2021) database. All patients were randomly separated into train cohort (*n* = 320) and test cohort (*n* = 217). Then, we summarized the clinical information including age at diagnosis, gender, OS, survival state, histological stage, pathological stage, and clinical stage, as shown in Table 1. Besides, the detailed clinical information of ccRCC patients in training cohort and test cohort were presented in Appendix A, respectivelyIn addition, the Tumor IMmune Estimation Resource (TIMER, https://cistrome.shinyapps.io/timer/, accessed on 2 July 2021) database was used to obtain the expression of genes in pan-cancer.

### 2.2. Establishment of a Protein–Protein Interaction Network

Firstly, we established a Protein–Protein Interaction Network (PPI) with STRING (https://string-db.org, accessed on 2 July 2021) Online Database. Then, Cytoscape software (V3.7.2, https://cytoscape.org, accessed on 2 July 2021) was adopted to analyze the interaction degree of collagen-related genes and re-visualize the interaction network. To screen hub genes of PPI, the plug-in named network analyzer was adopted to count nodes degree, which was characterized as the interaction degree.

### 2.3. Collagen-Related Risk Model Establishment

The univariate cox regression was performed to figure out the relationship between overall survival (OS) and hub collagen-related genes. Hub collagen-related genes screened with statistical significance were then embedded into multiple regression to acquire the coefficients, the risk score was acquired based on the following formula: risk score = $\overset{N}{\underset{i=1}{\sum }}\left(Expi*Coefi\right)$, where N = 5, Expi was the expression level of each collagen-related gene, Coefi was the corresponding coefficient obtained from multiple regression.

### 2.4. GSEA Analysis and Immune Cell Type Fraction Estimation

GSEA was used to investigate the discrepancy in the set of genes expressed between the high risk and low risk groups in the enrichment of the MSigDB Collection (h.all.v7.4.symbols.gmt; c5.go.v7.4.symbol.gmt). The risk scores were used as the phenotype label. All other parameters were default. CIBERSORT is an analytical tool from the Alizadeh Lab developed by Newman et al. to provide an estimation of the abundances of member cell types in a mixed cell population using gene expression data. This is also a publicly available R package that could be easily obtained from the website (https://cibersort.stanford.edu/, accessed on 2 July 2021). Here, CIBERSORT was adopted to assess the fractions of 22 immune cell types.

### 2.5. Statistics

We used the R language software (V.4.0.2, https://cran.r-project.org, accessed on 2 July 2021) to handle the analysis in this study. Wilcoxon rank sum test was used to compare the mRNA levels of collagen-related genes between ccRCC samples and normal samples. The hub genes in risk model were identified based on the univariate and multivariate cox regression. We adopted the ‘survival’ R package and the log-rank test to plot and assess the Kaplan–Meier (KM) curve. To verify the precision of our risk signature in forecasting the prognosis (OS) of ccRCC patients, ROC curves were generated using ‘survivalROC’ R package.

## 3. Results

### 3.1. Characteristics of Patients Separated into Training and Test Cohort

Clinical and pathological information of 537 ccRCC patients and 72 normal tissue samples were obtained from the cancer genome atlas (TCGA, https://portal.gdc.cancer.gov/, accessed on 2 July 2021) database. The sample function in the R platform was used to randomly separate all patients into a training cohort (*n* = 320) and test cohort (*n* = 217). The size parameter was set as 0.6 to extract 60% of all samples into the training cohort, while the replace parameter in this function was set as FALSE to ensure sampling without replacement. Then, we summarized the clinical information including age at diagnosis, gender, OS, survival state, histological stage, pathological stage, and clinical stage, as shown in Table 1. All variables were observed with no significant difference between the training cohort and the test cohort, indicating that the two samples had a high degree of consistency.

### 3.2. Establishment of a Collagen-Related Risk Model

The collagen-related gene set, which contained 257 genes (Appendix A), was adopted from the Molecular Signature Database (MsigDB, http://www.gsea-msigdb.org/gsea/msigdb/index.jsp, accessed on 2 July 2021). For better comprehension, we embedded 257 collagen-related genes into the STRING online database (https://string-db.org/, accessed on 2 July 2021) to construct a protein–protein interaction network. Based on the interaction degrees, the Cytoscape software was used to re-visualize the interaction network and screen out the top 25 hub genes with the highest degree (Figure 1A), suggesting their critical relationship with collagen.

We sought to construct a collagen-related risk model that can predict the prognosis of ccRCC patients using single and multiple stepwise regression analyses based on the top 25 collagen-related genes (10 percent of total genes) in the training cohort. In the single factor cox regression, 16 genes with statistical significance were correlated with patients’ OS (Appendix A). Subsequently, we performed a multivariate cox regression with these genes and generated a five collagen-related signature containing IL6, FN1, and three genes encoding collagen (COL4A4, COL9A2, COL7A1) to predict the prognosis of ccRCC patients (Figure 1B). Each patient in this study obtained a risk score, which was calculated with the following formula:Risk score = (0.0015 × FN1) + (0.019 × IL6) + (−0.1338 × COL4A4) + (0.0772 × COL9A2) + (0.0422 × COL7A1).

All five collagen-related genes were not significantly correlated with each other in both the training and test cohorts, indicating this risk model avoided the overfitting caused by collinearity (Figure 1C,D).

Then, we searched in the Tumor IMmune Estimation Resource (TIMER, https://cistrome.shinyapps.io/timer/, accessed on 2 July 2021) database to preview the expression of these five collagen-related genes in pan cancer. Figure 2A–E demonstrates that three collagen-related genes were found to be differently expressed in genitourinary tumors such as bladder cancer (BLCA), clear cell renal cell cancer (ccRCC/KIRC), and prostate cancer (PRAD). Moreover, the expression of FN1 was found to be significantly differently expressed in KIRC and PRAD, while IL6 was significantly differently expressed in BLCA, KIRC, and PRAD (*p* < 0.1 was considered statistically significant according to the TIMER database).

### 3.3. Prognostic Value of the Collagen-Related Signature in ccRCC Patients

Having developed the five collagen-related risk model, samples in the training cohort and test cohort were assigned a risk score. Then, the medium value of the risk scores in the training cohort was set as a cutoff to judge and separate patients into high risk and low risk groups in both the training and test cohort (Figure 3A,B).

As the main component of the extracellular matrix, collagen is closely related to the degree of tumor malignancy, whether it increases or decreases. Hence, the prognostic significance of the collagen-related signature was further detected. As demonstrated in the heatmap (Figure 3C,D), the expression level of COL4A4 was increased in the low risk group, while the expression levels of the remaining four genes were increased in the high risk group in both the train and test cohort. Compared to the low risk group, the fatality rate in the high risk group was significantly higher in both the training and testing cohort (Figure 3E,F). Furthermore, the implications for the prognosis of the collagen-related risk model in ccRCC were assessed with Kaplan–Meier analysis. According to Figure 3G, patients with high risk scores tend to obtain unfavorable overall survival in the training cohort, and this result was confirmed by the test cohort (Figure 3H).

### 3.4. The Expression of Collagen-Related Genes Is Associated with Clinical and Pathological Characteristics

In view of the significant biological function of collagen in the occurrence and development of tumors, we comprehensively analyzed the relationship between the five collagen-related genes and the clinicopathological characteristics of ccRCC, including clinical stage and WHO grades. In both the training and test cohort, the gene expression level of COL4A4 was increased in the low clinical stage groups, while the expression levels of COL7A1 and IL6 were increased in the high clinical stage groups (Figure 4A,B), implying that COL4A4 might act as a critical protective factor, while the other two genes are risk factors in tumor development. Quantitative analysis was also performed and confirmed a significant association between tumor stage and mRNA expression in the training (Figure 4C) and test (Figure 4D) cohorts. Furthermore, we investigated the association between gene expression and T, M stage, and WHO grade in the training cohort, separately. The findings revealed that the mRNA expression of COL4A4 was stably increased in low T and M stage groups as well as in the low WHO grade group (Figure 4E–G), and these results were further validated in the test cohort (Appendix A), implying that COL4A4 might act as an important predictive factor in tumor development.

### 3.5. The Accuracy of Collagen-Related Signature for Prognosis Evaluation

To estimate the accuracy of the overall survival prediction of the collagen-related signature, the received operating characteristic (ROC) curves were analyzed based on datasets from the training and test cohorts. The area under the ROC curve at 1 year and 5 years was up to 0.758 and 0.729, respectively, indicating a relatively high accuracy of prognosis prediction (Figure 5A). Results were subsequently confirmed in the test cohort (Figure 5B).

Moreover, the single and multiple stepwise regression analyses were adopted to estimate the independent prognostic value of the collagen-related signature. The single factor regression analysis revealed that patients with high risk scores were associated with unfavorable overall survival (Figure 5C) as well as other evaluating indicators including age, WHO grade, clinical stage, and pathological stage.

This was further validated in the test cohort (Figure 5D). Moreover, multiple stepwise regression analysis demonstrated that the high risk score tended to be independently correlated with unfavorable overall survival in ccRCC patients (Figure 5E), indicating an independent prognostic value for ccRCC. This was also validated in the test cohort (Figure 5F).

### 3.6. GSEA Identifies Potential Signaling Pathways

GSEA was adopted to investigate the potential signaling pathways activated in the high risk group. Genes were differently concentrated in high risk groups based on data from the training cohort, as they were related to tumorigenesis and the immune response, such as the epithelial–mesenchymal transition, inflammatory response, and IL6-JAK-STAT3 signaling pathway (Figure 6A). Similarly, this was also verified with the test cohort (Figure 6B).

### 3.7. Identification of Immune Cells Infiltrated in Patients with Different Risk Scores

As the immune response was identified as activated in the high risk group, we explored the possibility of a collagen-related signature in assessing the immune microenvironment. The CIBERSORT method was utilized to evaluate the discrepancies in the immune infiltration of 22 immune cell types between variable risk groups. Figure 7A summarizes the findings acquired from 320 ccRCC patients in the training cohort and 217 patients in the test cohort. Patients in the high risk group possessed a relatively higher ratio of regulatory T cells (Tregs) (Figure 7) but lower proportions of macrophages M1 (Figure 7C), indicating that patients with a high risk score might generate an immunosuppressive environment.

Moreover, GSEA was utilized to probe the connection between biological processes associated with immunity and the collagen-related signature. Results suggested that high risk ccRCC samples were remarkably correlated with negative regulation of the immunity pathway, such as negative regulation of the immune response, CD4+ αβ T cell activation, αβ T cell activation, and T cell-mediated immunity (Figure 7D).

Considering that patients with a high risk score were correlated with an immunosuppressive microenvironment, thus, collagen-related genes might be a critical target to improve the efficiency of immunotherapy.

### 3.8. The Immune Microenvironment of Patients with High Risk Scores Tend to Be Suppressed

As we know, the immune system initiates seven steps to eradicate tumor cells: (1) cancer cell antigen release, (2) cancer antigen presentation, (3) priming and activation, (4) convey T cells to tumors, (5) T cell infiltrate into tumors, (6) recognition of tumor cells, (7) tumor cell killing, which is defined as the tumor immunity cycle [24]. In this process, immune cells could be activated or suppressed through the combination of the ligand and receptor. For example, the function of CD8+T cells would be suppressed when PD-1 binds to PD-L1. Recently, increasing immune checkpoint inhibitors were invented to block the combination, thus restoring the function of immune cells [25,26]. Hence, in this work, we compared the expression of genes negatively regulating the processes in the cancer immunity cycle between the low risk and high risk cohorts. Gene symbols were obtained from the Tracking Tumor Immunophenotype website (http://biocc.hrbmu.edu.cn/TIP/, accessed on 2 July 2021). According to Figure 8A,B, the majority of genes including cytokines (IL10, TGFB1, etc.) and immune checkpoints (PD-1, CTLA4, etc.) that negatively regulate the tumor immune circulation were overexpressed in the high risk group, suggesting that the activity of the immune response in these patients was suppressed.

As our previous findings revealed that the proportions of Tregs are increased in the high risk group, the expression of molecules correlated with immune checkpoints were analyzed in both the low and high risk groups. Findings demonstrated that the expression of LAG3 and CTLA-4, which positively associated with the collagen-related risk score, were upregulated in the high risk group (Figure 8C–F). In addition, the mRNA level of other significant immune checkpoints such as PD1 and TIGIT were significantly upregulated in the high risk group (Figure 8G–H), while PD-L1 was downregulated in the high risk group (Figure 8I). As is well known, tumor immunosuppressive cytokines play a critical role in immune cell exhaustion. For instance, IL10, secreted by M2-macrophages, could impair the proliferation and migration of T cells [27]. Hence, we compared the expression of two immunosuppressive cytokines in the high and low risk groups. Results showed that IL10 and TGFB1 were both upregulated in the high risk group (Figure 8J), indicating that the immunosuppressive microenvironment might be further promoted through cytokines.

## 4. Discussion

Evidence from previous research showed that collagen-related genes are aberrantly expressed in variable tumors, such as bladder cancer and prostate cancer [28,29,30]. However, there is limited information about the expression and roles of collagen-related genes in ccRCC. In this study, we sorted out 257 collagen-related genes from the MSigDB and they are listed in Appendix A. Then, a PPI network was constructed and the top 10 percent of these genes with supreme degrees of interaction were screened out (Figure 1A). Subsequently, the single and multiple stepwise regression analyses were adopted to analyze the prognostic impact on ccRCC patients. Finally, a risk model that could predict ccRCC patients’ prognosis based on five hub collagen-related genes was constructed (Figure 1B). These findings promote the identification of novel biomarkers for the prediction of diagnosis and prognosis of ccRCC.

As was reported, several hub collagen-related genes including FN1, IL6, COL4A4 and COL7A1 were involved in the progression and development of variable cancers. Fibronectin 1 (FN1), an extracellular matrix glycoprotein, plays major roles in cell adhesion, migration, and differentiation [31,32]. Remodeling of this component has been shown to contribute to invasion and metastasis, while also acting in certain tumors as a physical barrier to block immune cell infiltration, thus resulting in poor immune therapy response. Interleukin-6 (IL-6), which plays a significant role in cancer progression, is a pleiotropic factor that belongs to a cytokine subfamily [33]. Evidence showed that IL-6 could mediate downregulation of type II collagen through the JAK/STAT pathway [34]. Furthermore, compared with COL VI- fibroblasts, preferential activation of the IL6/STAT3 pathway was observed in COL VI+ fibroblasts, indicating that IL6 contributed greatly to collagen regulation [35]. Additionally, research showed that IL6 were highly secreted in tumor cell/fibroblast co-cultures. Blockade of IL6 in tumors co-cultured with fibroblasts resulted not only in the regression of tumor growth but also in the accumulation of CD8+ TILs in intratumoral tissues [36]. Collagen Type IV Alpha 4 Chain (COL4A4) encodes one of the six subunits of type IV collagen, the major structural component of basement membranes. Mutations in this gene are associated with type II autosomal recessive Alport syndrome (hereditary glomeruli nephropathy) and with familial benign hematuria (thin basement membrane disease) [37]. Additionally, COL4A4 was found to be downregulated in esophageal tumor tissues [38,39]. COL7A1 gene encodes for collagen type VII, and was found aberrantly expressed in esophageal squamous cell carcinoma [40]. Moreover, high levels of type VII collagen expression were found to be correlated with the migration and invasion of recessive dystrophic epidermolysis bullosa cutaneous squamous cell carcinoma keratinocytes [41]. Nevertheless, these genes were rarely reported in ccRCC. In this research, COL4A4, COL7A1 and IL6 were found with extraordinary expression and were found to be associated with clinicopathological features with statistical significance. In contrast with IL6 and COL7A1, the lower expression of COL4A4 was often correlated with a lower clinical stage, pathological stage, and WHO grade, indicating COL4A4 is a protecting factor, while IL6 and COL7A1 are risk factors of ccRCC.

Moreover, based on the expression of five hub genes related with collagen, a risk model was established to forecast the prognosis of ccRCC patients in both training and testing groups; KM curves demonstrated that the high risk group was tremendously correlated with unfavorable OS (Figure 3G,H). Furthermore, ROC curves revealed that the signature of five collagen-related hub genes provide a significant value for ccRCC patients with unfavorable OS (Figure 5A,B).

Furthermore, to investigate whether the molecular biology mechanism of the five collagen-related genes promoted clear cell renal carcinoma genesis and progression, ccRCC patients in the training cohort were separated into high and low risk groups based on the median risk score. Results revealed that patients in the high risk group tend to correlate with epithelial–mesenchymal transition (EMT) and immune response (Figure 6). Recent studies have found that tumor epithelial cells and their adjacent normal epithelial cells can be transformed into cancer-associated fibroblasts (CAFs) through EMT, which can induce the invasion and migration of tumor cells and promote the development of tumors [42,43]. Activated CAFs can promote migration by secreting extracellular matrix components such as collagen glycotenin. Through expressing a series of growth factors and cytokines, VEGF, and monocyte chemotactic protein1 (MCP1), it can further activate the tumor matrix and promote the formation of a microenvironment needed for tumor development [44].

Despite several immune cell intrinsic mechanisms of immunotherapy resistance being described, the role of the extracellular matrix (ECM) in antitumor therapy remains not fully stated. Research showed that anti-PD-1/PD-L1 tumors were often correlated with increased collagen, and a reduction in LOX12-dependent collagen deposition could re-sensitize tumors to PD-1/PD-L1 blockade [45]. Moreover, another report demonstrated that high collagen density could instruct macrophages to acquire an immunosuppressive phenotype (TAMs), thus resulting in the low efficiency of attracting cytotoxic T cells and a higher ability to inhibit T cell proliferation [46]. Moreover, TAMs were also reported to regulate tumor sustained growth by secreting collagen I through activation of the prosurvival integrin α2β1/PI3K/AKT signaling pathway [47]. Hence, from this perspective, we compared the proportion of immune cells infiltrated in the high risk and low risk groups. Findings revealed that the content of regulatory T cells (Tregs) in high risk groups was relatively high, while the proportion of M1 macrophages that contribute to the antitumor response was relatively low (Figure 7B,C), indicating that the immune microenvironment of people with high risk scores are likely to be suppressed. Results of GSEA based on the gene ontology gene set also revealed that patients in high risk groups are negatively correlated with the regulation of immune pathways, for example, negative regulation of immune response, CD4 + αβ T cell activation, αβ T cell activation, and T cell-mediated immunity (Figure 7D). These results imply that our risk model might have the potential to predict the immune microenvironment.

As cytokines function as a critical role in the regulation of tumor immunity, we investigated the expression of two immunosuppressive cytokines including IL-10 and TGFB1. IL-10 helps maintain the expression of Foxp3 and TGF-β, thereby stabilizing the phenotype and function of Treg [48]. Moreover, TGF-β contributes to the inhibition of NK cell activity and dendritic cell maturation as well as the decrease in cytokine production [49,50]. Corresponding to these researches, the expression level of IL-10 and TGF-β were found to be overexpressed in the high risk group of patients (Figure 8J), which might further promote the immunosuppression.

Except for immunosuppressive cytokines, immune checkpoints also play an important part in tumorigenesis through the promotion of tumor immunosuppressive effects. For instance, tumors have the capability to stimulate immune checkpoint targets such as PD-L1 to bind with PD-1 expressed by immune cells, thus protecting themselves from being attacked. Although immune checkpoint inhibitors (ICIs) do work, many patients are still primarily resistant to ICIs. Thus, it is necessary to figure out the mechanism of immune checkpoint inhibitor resistance. Limited research has demonstrated that a high expression of collagen could exhaust the proportion of CD8+ T cells through activation of the LAIR1 receptor, which is upregulated following CD18 interaction with collagen. Targeting the impression of interaction between collagen and CD18 could enhance the sensitivity of anti-PD-1 to lung cancer [45]. However, the correlation between collagen and the immune checkpoint has not been fully stated. In our study, we analyzed the correlation between several immune checkpoints and risk scores between the high and low risk groups. Among them LAG-3 and CTLA-4 were found with relatively high positive correlation. Moreover, the expression of these genes was also found to be overexpressed in the high risk groups (Figure 8C–F).

Collectively, our risk model based on five collagen-related genes has a better prognostic value for ccRCC patients. Furthermore, our risk model could also predict an immunosuppressive microenvironment with high infiltration of regulatory T cells (Tregs) and low infiltration of M1-macrophages for ccRCC patients. Targeting immune checkpoints such as LAG-3 and CTLA-4 might contribute to the treatment of ccRCC. Nevertheless, there are several shortcomings worth mentioning. First, our risk signature was merely constructed based on data from the TCGA database and was only validated with internal data, which need to be further validated with external data and a clinical patient cohort as well as a multi-center study. Second, further investigations, including in vivo and in vitro experiments, are required to elucidate the molecular biological mechanisms.

## 5. Conclusions

To sum up, a prognostic risk signature consisting of five collagen-related genes was established, which were closely related with clinicopathology and immune response. Our results demonstrated a new perspective for the individualized diagnosis of ccRCC patients.

## Figures and Tables

**Figure 1 vaccines-09-01510-f001:**
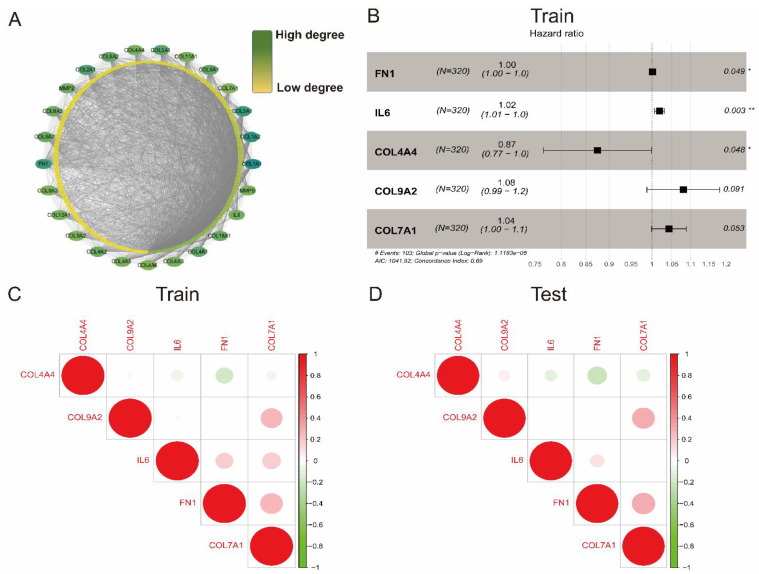
Identification of collagen-related signature to predict prognosis of ccRCC. (**A**) 25 hub genes dragged out from 257 collagen-related genes based on interaction degrees; (**B**) Establishment of a collagen-related risk model by univariate and multivariate cox regression; (**C**,**D**) Spearman correlation analysis of five collagen-related genes in train and test cohort. * *p* < 0.05, ** *p* < 0.01.

**Figure 2 vaccines-09-01510-f002:**
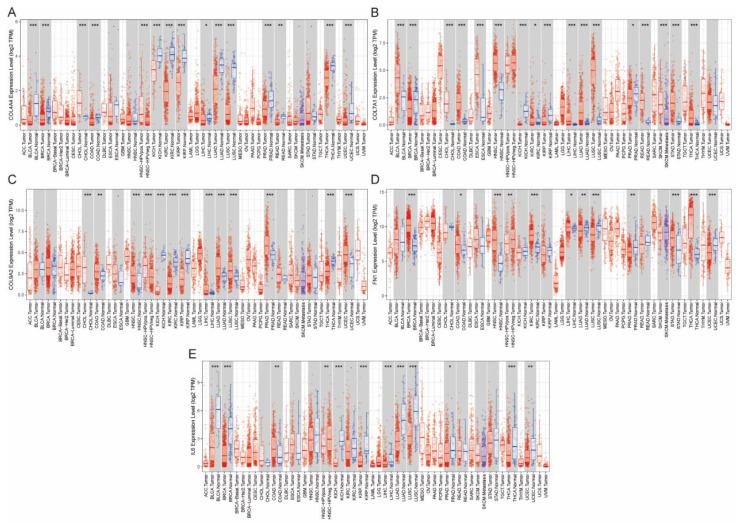
The gene expression profiles of five collagen-related genes in pan cancer based on TIMER database. (**A**–**E**) The differential expression between tumor and adjacent normal tissues for five collagen-related genes including COL4A4, COL7A1, COL9A2, FN1, and IL6 across all TCGA tumors. Distributions of gene expression levels are displayed using box plots, with statistical significance of differential expression evaluated using the Wilcoxon test. * *p* < 0.05, ** *p* < 0.01 and *** *p* < 0.001.

**Figure 3 vaccines-09-01510-f003:**
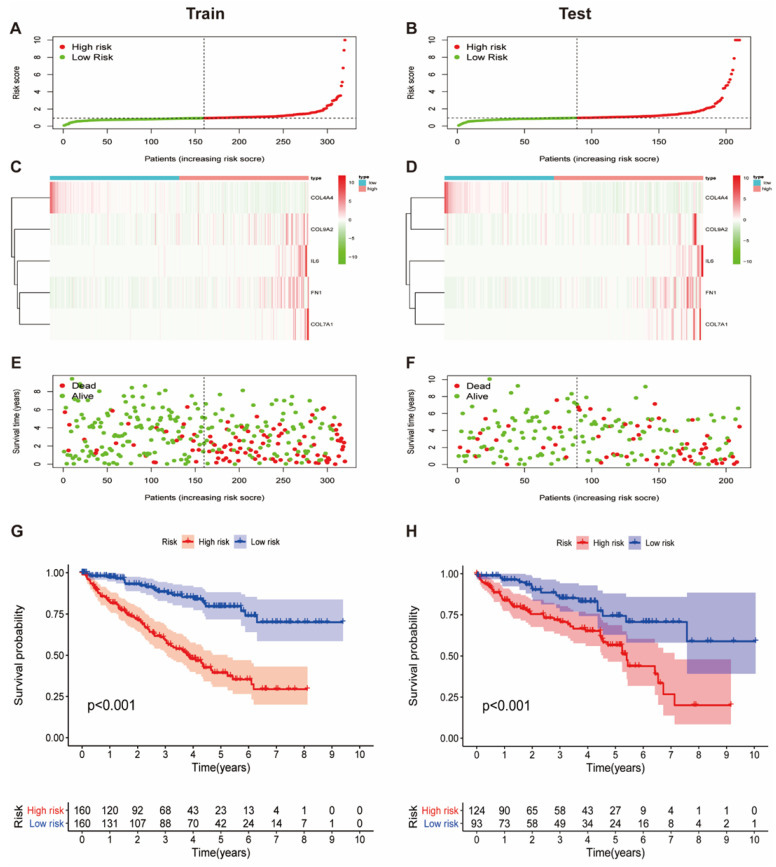
Prognosis value of collagen-related signature in train and test cohort. (**A**,**B**) Risk curve of ccRCC patients in train and test cohort. The median risk score in train cohort was set as the cutoff value to separate patients into high and low risk groups. The cutoff value in train cohort was also used to calculate the risk score of patients in test cohort; (**C**,**D**) The heatmap showing five hub gene expression profiles in high and low risk groups from train and test cohort; (**E**,**F**) Patient status distribution in high and low risk groups; (**G**,**H**) The Kaplan–Meier overall survival curves for patients assigned to high and low risk groups based on the risk score.

**Figure 4 vaccines-09-01510-f004:**
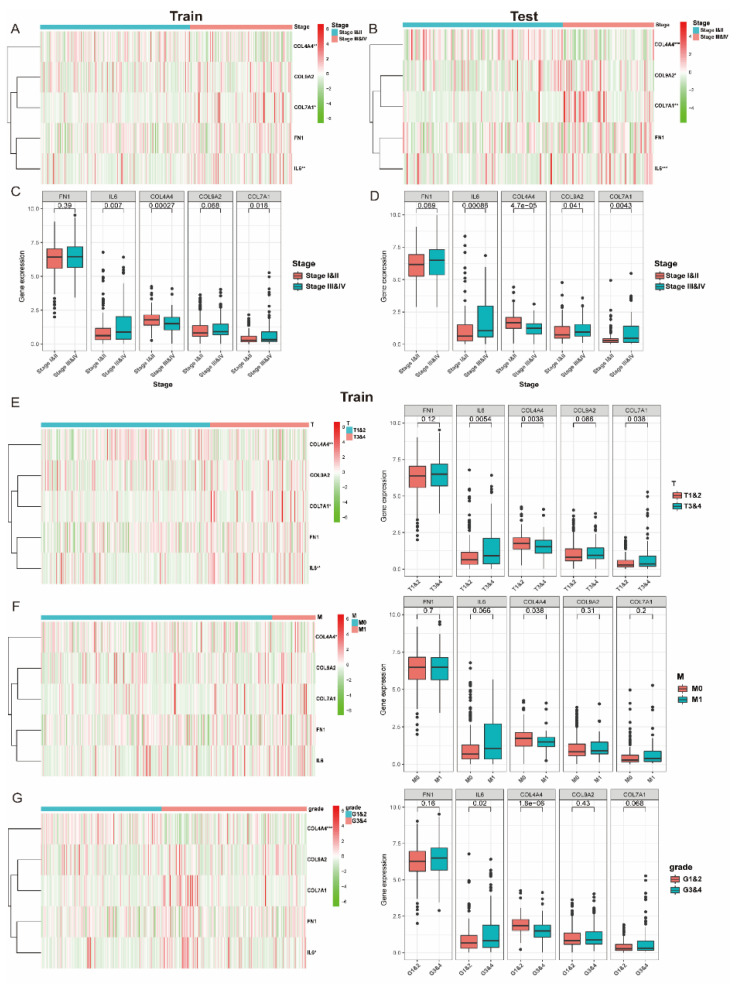
Collagen-related gene expression is correlated with clinicopathological features of ccRCC. (**A**,**B**) The heatmap showing five collagen-related gene expression profiles in different clinical stages from train and test cohorts; (**C**,**D**) The expression levels of five collagen-related genes in ccRCC with different clinical stages; (**E**–**G**) The heatmap and expression levels of five collagen-related genes in different T stage, M stage, and WHO grades from train and test cohorts; * *p* < 0.05,** *p* < 0.01 and *** *p* < 0.001.

**Figure 5 vaccines-09-01510-f005:**
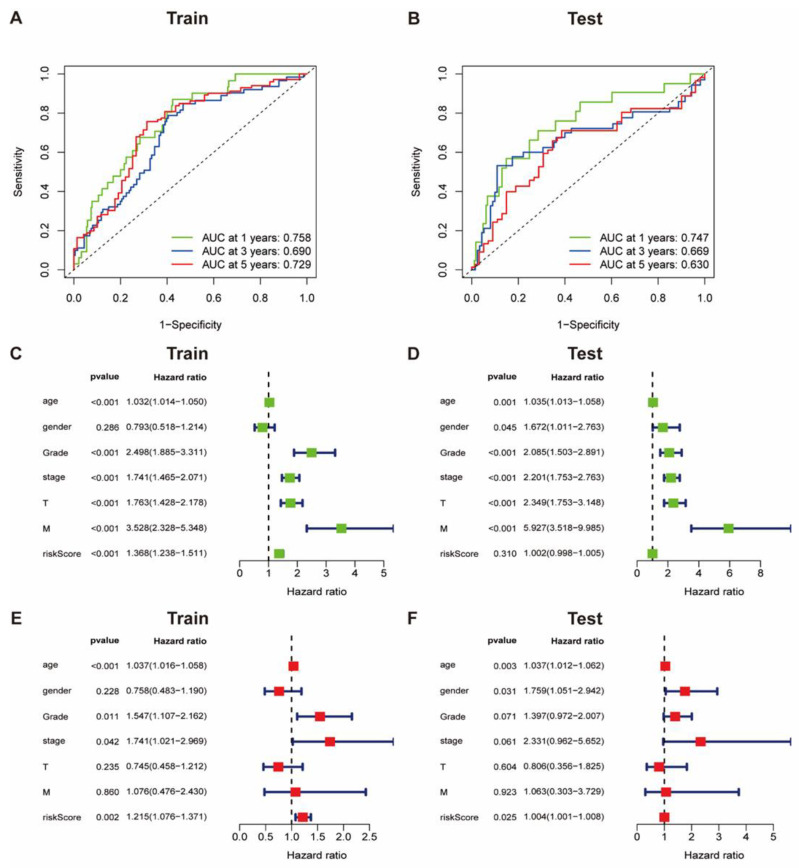
Prognostic value of the collagen-related signature in ccRCC. (**A**,**B**) ROC curves showing the predictive efficiency of the collagen-related signature on the 1-,3- and 5-years survival rate; (**C**–**F**) Univariate and multivariate cox regression analysis evaluating the independent prognostic value of collagen-related signature in terms of OS in ccRCC patients.

**Figure 6 vaccines-09-01510-f006:**
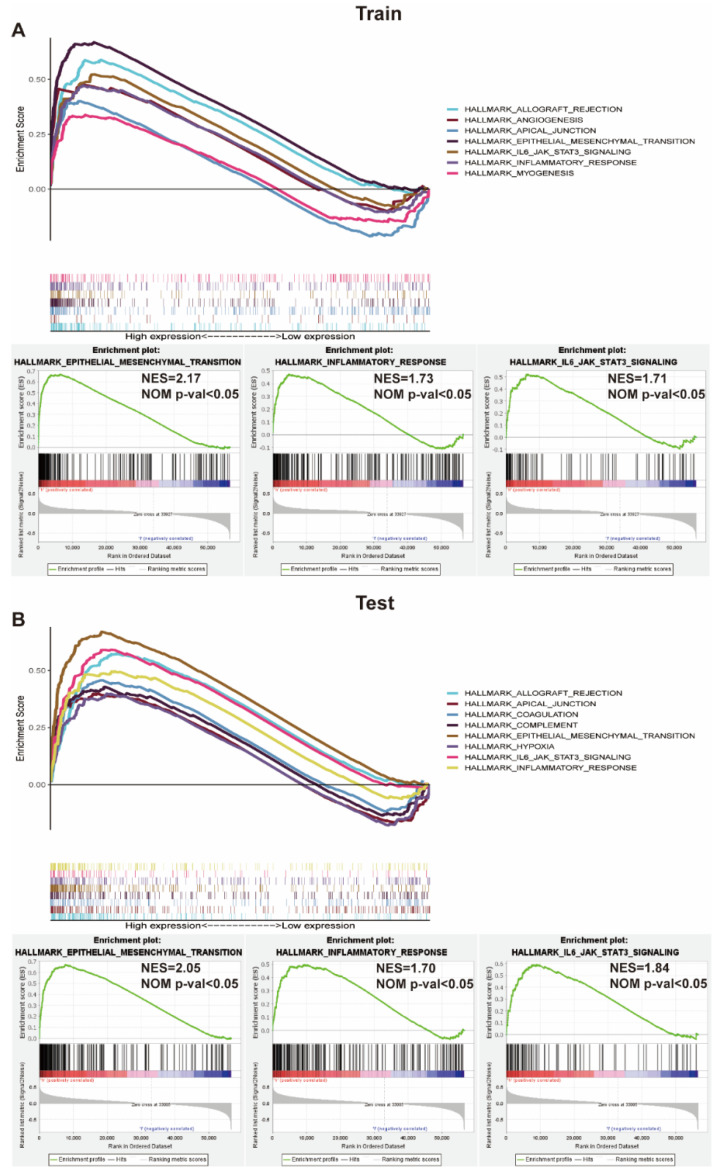
GSEA enrichment analysis between high and low risk groups. (**A**) The hallmark enrichment of high and low risk groups by GSEA method; GSEA revealing that genes in the high risk group were enriched for hallmarks of epithelial–mesenchymal transition (EMT) and immune response in train cohort; (**B**) The results were further validated by the test cohort.

**Figure 7 vaccines-09-01510-f007:**
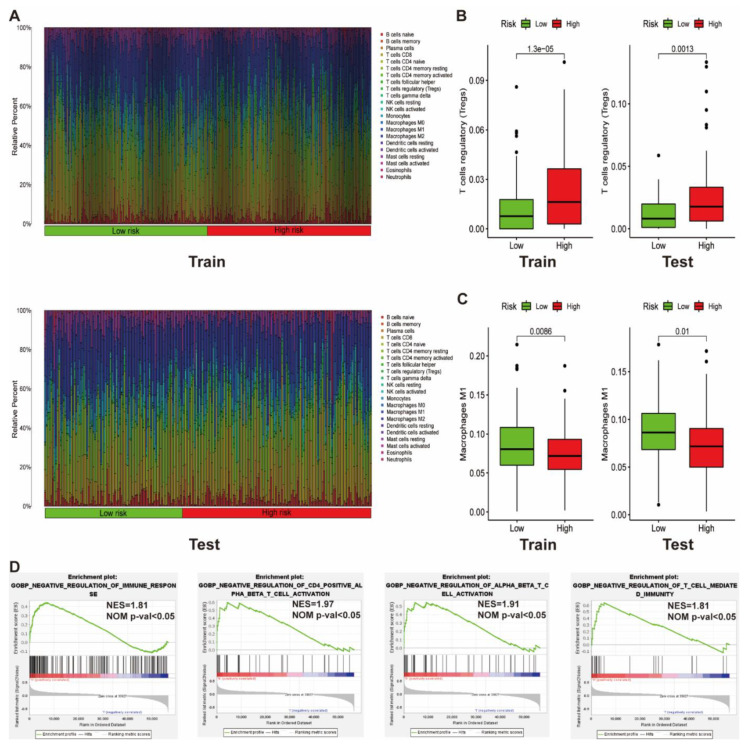
Immune landscape between low and high risk ccRCC patients. (**A**) Relative proportion of immune infiltration in high and low risk patients. (**B**,**C**) Box plots visualizing significantly different immune cells between high risk and low risk patients. (**D**) GSEA demonstrating that collagen-related signature correlated with immune-related biological function.

**Figure 8 vaccines-09-01510-f008:**
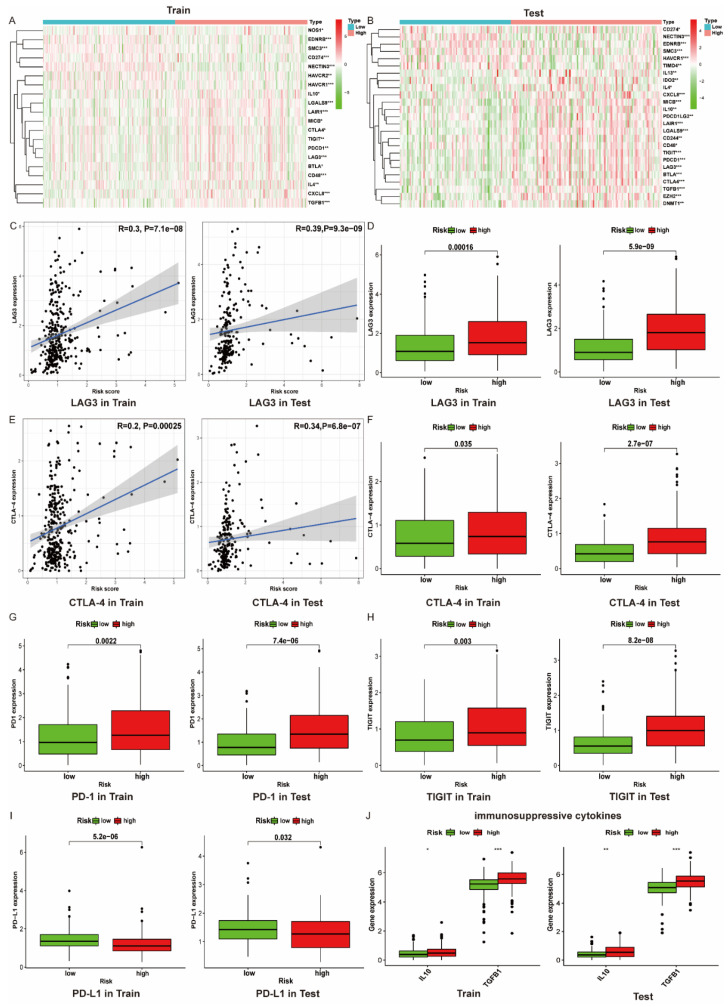
High collagen-related risk score indicates an immunosuppressive microenvironment. (**A**,**B**) Heatmap of gene profiles involved in the negative regulation of the cancer immunity cycle in high and low risk groups in train and test cohorts; (**C**) Correlation between LAG3 expression and risk score; (**D**) LAG3 expression in high and low risk groups; (**E**) Correlation between CTLA-4 expression and risk score; (**F**) CTLA-4 expression in high and low risk groups; (**G**–**I**) PD1,TIGIT, and PD-L1 expression in high and low risk groups; (**J**) Tumor immunosuppressive cytokines expression in high and low risk groups; * *p* < 0.05,** *p* < 0.01 and *** *p* < 0.001.

**Table 1 vaccines-09-01510-t001:** Clinical characteristics of the ccRCC patients in Train and Test cohorts.

Variables	Training Cohort (*n* = 320)	Test Cohort (*n* = 217)	*p* Value
**Age**			
Mean (SD)	60.03 (11.81)	61.42 (12.63)	0.194
Median (min,max)	60.00 (29,90)	61 (26,90)	
**Gender**			
MALE	209 (65.31%)	137 (63.13%)	0.605
FEMALE	111 (34.69%)	80 (36.87%)	
**Overall Survival time**			
Mean (SD)	1130.68 (803.79)	1146.57 (831.64)	0.824
Median (min,max)	1026.50 (3,3431)	1106.00 (2,3668)	
Unknown	3	2	
**Survival State**			
Alive	217 (67.81%)	150 (69.12%)	0.748
Dead	103 (32.19%)	67 (30.88%)	
**Histologic grade**			
G1	10 (3.13%)	4 (1.84%)	0.822
G2	133 (41.56%)	97 (44.70%)	
G3	127 (39.69%)	80 (36.87%)	
G4	45 (14.06%)	33 (15.21%)	
Unknown	5 (1.56%)	3 (1.38%)	
**T stage**			
T1	159 (49.69%)	116 (53.46%)	0.851
T2	43 (13.44%)	26 (11.98%)	
T3	111 (34.69%)	71 (32.72%)	
T4	7 (2.19%)	4 (1.84%)	
Unknown	0 (0%)	0 (0%)	
**N stage**			
N1	12 (3.75%)	5 (2.30%)	0.221
N0	134 (41.88%)	106 (48.85%)	
Unknown	174 (54.38%)	106 (48.85%)	
**M stage**			
M1	48 (15.00%)	31 (14.29%)	0.519
M0	256 (80.00%)	170 (78.34%)	
Unknown	16 (5.00%)	16 (7.37%)	
**Clinical stage**			
Stage I	155 (48.44%)	114 (52.53%)	0.8803
Stage II	34 (10.63%)	23 (10.60%)	
Stage III	79 (24.69%)	46 (21.20%)	
Stage IV	50 (15.63%)	33 (15.21%)	
Unknown	2 (0.63%)	1 (0.46%)	

## Data Availability

The datasets analysed during the current study are available in the (TCGA and STRING) repository, (https://portal.gdc.cancer.gov/; https://string-db.org, accessed on 2 July 2021).

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
