# Peer review of "A Five Collagen-Related Gene Signature to Estimate the Prognosis and Immune Microenvironment in Clear Cell Renal Cell Cancer"

_vaccines, 2021, doi:10.3390/vaccines9121510_

Round 1
Reviewer 1 Report
The paper by Shi et al. is an article whose aim was to establish a collagen-related signature to predict the prognosis and estimate the tumor immune microenvironment in clear cell renal cell cancer patients. After sorting out many collagen-related genes, the Authors highlighted five of them to predict the outcome of this type of cancer in correlation with clinical pathological features.
I think the paper is altogether clear and well structured in the frame of the examined subject and title. The title clearly and precisely reflect the findings of the manuscript.
The introduction could explain more comprehensively the clinical background, on the other hand the Results were appropriately described and well argued.
Nevertheless, there are some minor concerns which should be addressed by the Authors before a publication on Vaccines can be granted:
MINOR CONCERNS:
- in the “Introduction” paragraph, something more about collagen and its importance in tumor development should be added (at this regard, the Authors should cyte some important papers which treat this topic:
Cancer Metastasis Rev. 2020 Sep;39(3):603-623. Collagen biology making inroads into prognosis and treatment of cancer progression and metastasis. Ana C Martins Cavaco, Sara Dâmaso, Sandra Casimiro, Luís Costa;
Int J Oncol. 2019 Aug;55(2):391-404. COL6A1 promotes metastasis and predicts poor prognosis in patients with pancreatic cancer. Kwabena Gyabaah Owusu-Ansah, Guangyuan Song, Ronggao Chen, Muhammad Ibrahim Alhadi Edoo, Jun Li, Bingjie Chen, Jian Wu, Lin Zhou, Haiyang Xie, Donghai Jiang, Shusen Zheng ;
Cell Mol Bioeng. 2017 Mar 6;10(3):223-234. Investigating the Mechanobiology of Cancer Cell-ECM Interaction Through Collagen-Based 3D Scaffolds. Chiara Liverani, Laura Mercatali, Luca Cristofolini, Emanuele Giordano, Silvia Minardi, Giovanna Della Porta, Alessandro De Vita, Giacomo Miserocchi, Chiara Spadazzi, Ennio Tasciotti, Dino Amadori, Toni Ibrahim.)
- in “Results” paragraph, line 113, the Authors should explain the difference between train cohort and test cohort;
- in Table 1, treatments received by patients are missing;
-in “Results” paragraph, line 142: what are “FRGs”? Explain it;
- typing errors should be checked. Moreover, some sentences don’t make sense and the Authors need to correct them (check line: 33, 266-267,314, 341).
Overall, this is my final decision: ACCEPT AFTER MINOR REVISION.
Author Response
Response to Reviewer 1 Comments
Point 1:
The paper by Shi et al. is an article whose aim was to establish a collagen-related signature to predict the prognosis and estimate the tumor immune microenvironment in clear cell renal cell cancer patients. After sorting out many collagen-related genes, the Authors highlighted five of them to predict the outcome of this type of cancer in correlation with clinical pathological features.
I think the paper is altogether clear and well structured in the frame of the examined subject and title. The title clearly and precisely reflect the findings of the manuscript.
The introduction could explain more comprehensively the clinical background, on the other hand the Results were appropriately described and well argued.
Nevertheless, there are some minor concerns which should be addressed by the Authors before a publication on Vaccines can be granted:
Response 1:
Thank you very much for your kindness and careful revision on our manuscript!
Based on your kindness suggestion, we modified the manuscript which could be observed in red in the revised manuscript.
Point 2:
- in the “Introduction” paragraph, something more about collagen and its importance in tumor development should be added (at this regard, the Authors should cyte some important papers which treat this topic:
Cancer Metastasis Rev. 2020 Sep;39(3):603-623. Collagen biology making inroads into prognosis and treatment of cancer progression and metastasis. Ana C Martins Cavaco, Sara Dâmaso, Sandra Casimiro, Luís Costa;
Int J Oncol. 2019 Aug;55(2):391-404. COL6A1 promotes metastasis and predicts poor prognosis in patients with pancreatic cancer. Kwabena Gyabaah Owusu-Ansah, Guangyuan Song, Ronggao Chen, Muhammad Ibrahim Alhadi Edoo, Jun Li, Bingjie Chen, Jian Wu, Lin Zhou, Haiyang Xie, Donghai Jiang, Shusen Zheng;
Cell Mol Bioeng. 2017 Mar 6;10(3):223-234. Investigating the Mechanobiology of Cancer Cell-ECM Interaction Through Collagen-Based 3D Scaffolds. Chiara Liverani, Laura Mercatali, Luca Cristofolini, Emanuele Giordano, Silvia Minardi, Giovanna Della Porta, Alessandro De Vita, Giacomo Miserocchi, Chiara Spadazzi, Ennio Tasciotti, Dino Amadori, Toni Ibrahim.)
Response 2:
Thanks greatly for your suggestion for the introduction section.
After reading your suggestion, we found it is quite necessary for us to add more introduction about the importance of collagen in tumor development. The references suggested by you are quite interesting and helpful. By the way, these references were cited in this paper.
The introduction was modified in the revised manuscript. For your convenience, we presented the revised section in the following:
Revised introduction:
As is known to all, the seed-soil theory was proposed to elucidate the development process of tumors which is a dynamic process of interaction between tumor cells and their microenvironment [6–8]. Tumor microenvironment is considered as a heterogeneous tissue include not only tumor cells, but also immune cells, fibroblast, and endothelial cells. Besides, the noncellular component extracellular matrix (ECM) also play a critical role in this environment. In particular, the collagenous matrix is subjected to significant alterations in its composition and structure that create a permissive environment for tumor growth, invasion, and dissemination [9]. As the major com-ponent of ECM, collagen function as the attachment and scaffold for cell growth, which not only induces the proliferation, differentiation, and migration of epithelial cells, but also plays an important role in maintaining intercellular adhesion, tissue integrity and repairing as well as supporting organs [10]. Recently, increasing evidence showed that it plays a critical role in cancer progression and metastasis. For instance, collagen structural parameters (density, length, and width) and the presence of im-mature collagen molecules were reported to be associated with unfavorable out-comes of gastric cancer [11]. Furthermore, research demonstrated that excessive col-lagen deposition could reshape tumor environment thus define cell fate and promote tumor progression. Previously, the deposition of collagen was considered mainly re-leased by Cancer-associated fibroblasts (CAFs) [12]. However, increasing evidence showed that collagen could also be produced by tumor cells in a specific manner. For example, high expression level of collagen XIII was observed being produced in triple negative breast cancer cell lines [13]. These phenomena (collagen could be produced by cancer cells) were also observed in lung adenocarcinoma, esophageal squamous cell carcinoma cells and so on [14,15]. Furthermore, evidence demonstrated that density and structural alteration of collagen I fiber could be function as highways of cancer cells and macrophages for rapid travelling, thus results in cancer progression and metastasis [16–19].
Besides, accumulating evidence showed that collagen expression could serve as prognostic indicators of cancer progression and metastasis. Overexpression of COL11A1 was reported to be associated with relapse of non-small cell lung cancer [20]. In addition, upregulated COL6A1 in clear renal cell carcinoma and pancreatic cancer was also reported to be a predictive prognostic biomarker [21,22]. Moreover, in Head and neck squamous cell carcinoma, collagen was considered correlated with the alleviation of apoptotic response to cisplatin via discoidin domain receptor 1 (DDR1) [23].
Point 3:
- in “Results” paragraph, line 113, the Authors should explain the difference between train cohort and test cohort;
Response 3:
Thank you very much for your suggestion.
Based on your suggestion, the difference of these variables between training and test cohort were compared. P<0.05 was considered as statistic significant. We added the explanation from line 143 to 145. For your convenience, we presented the statement as following:
All variables, were observed with no significant difference between Training cohort and Test cohort, indicating that the two samples have a high degree of consistency.
Point 4:
- in Table 1, treatments received by patients are missing.
Response 4:
Thank you very much for your suggestion.
Indeed, it is necessary to discuss the overall survival on the basis of treatment options. However, the treatment received by patients are not well recorded in the TCGA database (KIRC). Hence, in this study, the treatment was not mentioned.
Thank you for your revision.
Point 5:
-in “Results” paragraph, line 142: what are “FRGs”? Explain it.
Response 5:
Thank you for your careful revision.
We are sorry for making a such mistake. FRGs were referred to ferroptosis-related genes which should be mentioned in another article. Here, in this manuscript, FRGs should be altered with collagen-related genes.
Thanks greatly for your kindness suggestion and careful revision!
Point 6:
- typing errors should be checked. Moreover, some sentences don’t make sense and the Authors need to correct them (check line: 33, 266-267,314, 341).
Response 6:
Thank you for your careful revision
Based on your suggestion, we checked the typing errors and non-sense sentences (Revised line: 315,366-368,402-403)
Thank you for your suggestions
To sum up, all response to reviewer 1 were above. Thanks greatly for taking time review our paper. We appreciate a lot for your contribution and constructive suggestions.

Reviewer 2 Report
Ref: vaccines-1495533
Title: A five collagen-related gene signature to estimate the prognosis and immune microenvironment in clear cell renal cell cancer
Journal: Vaccines
The manuscript entitled: “A five collagen-related gene signature to estimate the prognosis and immune microenvironment in clear cell renal cell cancer” by Shi et al. is a well written article related to the establishment of a five collagen-related gene signature to estimate the immune microenvironment and predict the prognosis of clear cell renal cell cancer patients. The manuscript could be considered for publication in the Vaccines journal, after some major modifications. Please find below the comments-suggestions that could be added to further improve the article below:
Specific major/minor comments:
Introduction:
-Line 57: “Collagen is the main component of ECM”: please specify the types of cancers. Also it would be better to describe first the composition of the tumor microenvironment and then mention the % on average of collagen and the importance related to the other component in the tumor microenvironment.
-A detailed description of the mechanism of collagen in cancer progression and possibly chemoresistance would be nice to be added here.
-General comment: There are a few typos or grammatical errors throughout the text; please go through and amend.
-Line 65: A prognostic biomarker is mainly for prognosis so maybe would be better to rephrase as “a biomarker to assist the diagnosis and improve the prediction of prognosis”
-Line 67:” ccRCC groups”: this is a very general term, please define and specify what type of groups.
Materials and Methods:
-Line 71: "RNA sequence" rephrase to “RNA sequencing”
-Line 78: “All parameters in this section were default”: please define those parameters.
- CIBERSORT: please state whether this is a publicly available software.
-General: Methodology could have been explained with more details
Results:
-Figure 2: Please include in the figure legend more details and briefly include the statistics that have been used.
-Please explain more clearly the discrimination between "train" and "test" cohort and why this was defined as random.
-Line 236: “22 immune cell types between variable risk groups.”: please explain the choice of these 22 immune cell types in terms of the immunosuppression profiling of the tumor.
-Line 246-247:” Consequently, to improve the efficiency of immunotherapy, collagen-related genes might be a critical target to focus on.”: please explain this sentence a bit more by associating your results with details.
-Line 252:” which negatively regulate the tumor immune circulation”: please expand and explain this a bit better.
-Line 256:” the expression of molecules correlated with immune checkpoints”: please link better the study of the immune checkpoints with the immune microenvironment and the significance to cancer immunotherapy.
-Figure 8: needs to be explained more thoroughly throughout the text. For instance which is the role of the immunosuppressive cytokines?
Discussion:
-Line 266: “aberrantly expressed in variable tumors”: please mention a few common tumor types.
-Lines 275-276: Please explain more clearly and with details how the collagen-related genes suggested impact collagen availability and action within the tumor microenvironment and how this may have an impact in tumor progression.
-Explain clearly how IL-6 which is a cytokine can be considered as collagen-related gene: which is the mechanism of action?
-Please whenever you mention your findings in the discussion, make a reference of the figures included in the manuscript.
-Lines 328-340: please explain more clearly by including more references to justify the correlation of the immunosuppression with the collagen-related genes based on your findings. Also you should discuss more clearly the impact your findings may have in cancer immunotherapy.
-The study of the immunosuppressive cytokines is not directly linked with your previous study in immunosuppression and the risk model. Please elaborate.
-Line 350-351: “Tumors have the capability to stimulate immune checkpoint targets such as LAG-3, CTLA-4, PD1, PD-L1, TIGIT and TIM-3 to protect themselves from attacking”: please explain better the mechanism of the immune checkpoints in relation to the immune/tumor cells interactions and support a link to immunotherapy.
-Line 360-361: “could also predict the immune environment of these patients”: this is a very general term you should specify the impact of the immune microenvironment related to cancer progression and resistance to therapy. Please elaborate.
Author Response
Response to Reviewer 2 Comments
Point 1:
The manuscript entitled: “A five collagen-related gene signature to estimate the prognosis and immune microenvironment in clear cell renal cell cancer” by Shi et al. is a well written article related to the establishment of a five collagen-related gene signature to estimate the immune microenvironment and predict the prognosis of clear cell renal cell cancer patients. The manuscript could be considered for publication in the Vaccines journal, after some major modifications. Please find below the comments-suggestions that could be added to further improve the article below:
Response 1:
Thank you very much for your kindness and careful revision on our manuscript.
Based on your kindness suggestions, we modified the manuscript which could be observed in blue in the revised manuscript.
Point 2:
Introduction:
-Line 57: “Collagen is the main component of ECM”: please specify the types of cancers. Also, it would be better to describe first the composition of the tumor microenvironment and then mention the % on average of collagen and the importance related to the other component in the tumor microenvironment.
-A detailed description of the mechanism of collagen in cancer progression and possibly chemoresistance would be nice to be added here.
-General comment: There are a few typos or grammatical errors throughout the text; please go through and amend.
Response 2:
Thanks greatly for your so much detailed suggestion. This really a great help for us to improve the quality of our work.
Based on your suggestion, we modified the introduction which was presented as following:
-Line 50-55: Tumor microenvironment is considered as a heterogeneous tissue include not only tumor cells, but also immune cells, fibroblast, and endothelial cells. Besides, the noncellular component extracellular matrix (ECM) also play a critical role in this environment. In particular, the collagenous matrix is subjected to significant alterations in its composition and structure that create a permissive environment for tumor growth, invasion, and dissemination [9]. As the major com-ponent of ECM, collagen function as the attachment and scaffold for cell growth, which not only induces the proliferation, differentiation, and migration of epithelial cells, but also plays an important role in maintaining intercellular adhesion, tissue integrity and repairing as well as sup-porting organs. Recently, increasing evidence showed that it plays a critical role in cancer progression and metastasis. For instance, collagen structural parameters (density, length, and width) and the presence of im-mature collagen molecules were re-ported to be associated with unfavorable outcomes of gastric cancer [10]. Furthermore, research demonstrated that excessive collagen deposition could reshape tumor environment thus define cell fate and promote tumor progression. Previously, the deposition of collagen was considered mainly released by Cancer-associated fibroblasts (CAFs) [11]. However, increasing evidence showed that collagen could also be pro-duced by tumor cells in a specific manner. For example, high expression level of collagen XIII was observed being produced in triple negative breast cancer cell lines [12]. These phenomena (collagen could be produced by cancer cells) were also observed in lung adenocarcinoma, esophageal squamous cell carcinoma cells and so on [13,14]. Furthermore, evidence demonstrated that density and structural alteration of collagen I fiber could be function as highways of cancer cells and macrophages for rapid travelling, thus results in cancer progression and metastasis [15–18].
Besides, accumulating evidence showed that collagen expression could serve as prognostic indicators of cancer progression and metastasis. Overexpression of COL11A1 was reported to be associated with relapse of non-small cell lung cancer [19]. In addition, upregulated COL6A1 in clear renal cell carcinoma and pancreatic cancer was also reported to be a predictive prognostic biomarker [20,21]. Moreover, in Head and neck squamous cell carcinoma, collagen was considered correlated with the alleviation of apoptotic response to cisplatin via discoidin domain receptor 1 (DDR1) [22].
Thank you very much for your revision.
Point 3:
-Line 65: A prognostic biomarker is mainly for prognosis so maybe would be better to rephrase as “a biomarker to assist the diagnosis and improve the prediction of prognosis”
Response 3:
Thank you very much for your precise and accurate revision.
We thought the suggestion is rational, thus we rephrased the sentence. The whole sentence was altered as following:
-Line 81-83: Hence, in this study, collagen-related genes were comprehensively sorted out to establish a gene signature which might serve as a prognostic biomarker to assist the diagnosis and improve the prediction of prognosis in clear cell renal cell cancer.
Point 4:
-Line 67:” ccRCC groups”: this is a very general term, please define and specify what type of groups.
Response 4:
Thank you very much for your revision.
We thought it might be a misunderstanding, due to the poor English level of us.
Here, all patients with clear cell renal cell carcinoma (ccRCC), including local and advanced ccRCC patients in the TCGA were including in this work.
So, we changed the phrase “groups” into patients in case of misunderstand. The whole sentence was presented as following:
-Line 83-84:This risk model might also be used to predict the immune microenvironment in ccRCC patients.
Materials and Methods:
Point 5:
-Line 71: "RNA sequence" rephrase to “RNA sequencing”
Response 5:
Thank you very much for your careful revision. We are very sorry for making such mistakes. Here, presented the whole sentence.
-Line 90-92:The transcriptome RNA sequencing data (FPKM value) and corresponding clinical and pathological information of 537 ccRCC patients and 72 normal tissue samples can be possessed from the TCGA (https://portal.gdc.cancer.gov/) database.
Point 6:
-Line 78: “All parameters in this section were default”: please define those parameters.
Response 6:
Thank you very much for your revision.
Actually, “gene symbol” is the only parameter which should be put in the Diff Exp module in TIMER database. We add this sentence only aim to explain that all parameters were set default by the website. In case of misleading, this sentence was removed.
Thanks greatly for your careful revision.
Point 7:
- CIBERSORT: please state whether this is a publicly available software.
Response 7:
Thank you very much for your suggestion.
CIBERSORT is an analytical tool from the Alizadeh Lab developed by Newman et al. to provide an estimation of the abundances of member cell types in a mixed cell population, using gene expression data. This is also a publicly available R package which could be easily obtained from the website (https://cibersort.stanford.edu/).
Thank you very much. We also added the statement in the manuscript. The paragraph was presented as following:
-Line 117-122: CIBERSORT is an analytical tool from the Alizadeh Lab developed by Newman et al. to provide an estimation of the abundances of member cell types in a mixed cell population, using gene expression data. This is also a publicly available R package which could be easily obtained from the website (https://cibersort.stanford.edu/). Here, the CIBERSORT was adopted to assess the fractions of 22 immune cell types.
Thank you very much for your suggestions.
Point 8
-General: Methodology could have been explained with more details
Response 8:
Thank you very much for your suggestion.
Based on your suggestion, the methodology was explained with more details which could be visualized in the manuscript in blue.
Thanks greatly for your kindness suggestion.
Results:
Point 9:
-Figure 2: Please include in the figure legend more details and briefly include the statistics that have been used.
Response 9:
Thank you very much for your suggestion.
We specified the figure legend of Figure 2 as following:
-Line 165-168: Figure 2. The gene expression profiles of five collagen-related genes in pan-cancer based on TIMER database. (A-E) The differential expression between tumor and adjacent normal tissues for five collagen related genes including COL4A4, COL7A1, COL9A2, FN1 and IL6 across all TCGA tumors. Distributions of gene expression levels are displayed using box plots, with statistical significance of differential expression evaluated using the Wilcoxon test. *P<0.05, **P<0.01 and ***P<0.001.
Thank you very much for your helpful and precise suggestion
Point 10:
-Please explain more clearly the discrimination between "train" and "test" cohort and why this was defined as random.
Response 10:
Thank you very much for your precise suggestion. The explanation was presented as following:
-Line 136-142: The sample function in the R platform was used to randomly separate all patients into training cohort (n=320) and test cohort (n=217). The ‘size’ parameter was set as 0.6 to extract 60% of all samples into training cohort while ‘replace’ parameter in this function was set as FASLE to ensure sampling without replacement.
Thanks greatly for your revision!
Point 11:
-Line 236: “22 immune cell types between variable risk groups.”: please explain the choice of these 22 immune cell types in terms of the immunosuppression profiling of the tumor.
Response 11:
Thank you very much for you question.
Admittedly, immunosuppressive cells include Tregs, TAMs, MDSC, CAFs and so on. However, in CIBERSORT, MDSC and CAFs could not be distinguished. So, in this study we selected Tregs and TAMs to make further analysis.
Thank you for your question.
Point 12:
-Line 246-247:” Consequently, to improve the efficiency of immunotherapy, collagen-related genes might be a critical target to focus on.”: please explain this sentence a bit more by associating your results with details.
Response 12:
Thank you for your revision
Based on your suggestion, we made the modification as following:
-Line 275-277: Considering that patients with high-risk score were correlated with immunosuppressive microenvironment. Thus, collagen-related genes might be a critical target to improve the efficiency of immunotherapy.
Thank you very much for your suggestion!
Point 13:
-Line 252:” which negatively regulate the tumor immune circulation”: please expand and explain this a bit better.
Response 13:
Thank you for your suggestion.
In this sentence, we are about to illustrate the source of genes visualized in the heatmap. We thought you might want us to explain the negative regulation of tumor immune circulation. So, we added the explanation of tumor immune circulation at the beginning of this paragraph:
-Line 280-287: As we known that immune system would initiate 7 steps to eradicate tumor cells: (1) cancer cell antigen release, (2) caner antigen presentation, (3) priming and activa-tion, (4) convey T cells to tumors, (5) T cell infiltrate into tumors, (6) recognition of tumor cells, (7) Tumor cell killing, which was defined as tumor immunity cycle
Thank you very much. We appreciate a lot for your constructive suggestion.
Point 14:
-Line 256:” the expression of molecules correlated with immune checkpoints”: please link better the study of the immune checkpoints with the immune microenvironment and the significance to cancer immunotherapy.
Response 14:
Thank you very much for your suggestion.
Here, we are about to state the molecules correlated with immune checkpoints. So, based on your suggestion, we added the explanation at the beginning of this section.
-Line 280-287: As we known that immune system would initiate 7 steps to eradicate tumor cells: (1) cancer cell antigen release, (2) caner antigen presentation, (3) priming and activation, (4) convey T cells to tumors, (5) T cell infiltrate into tumors, (6) recognition of tumor cells, (7) Tumor cell killing, which was defined as tumor immunity circulation. In this process, immune cells could be activated or suppressed through combination of ligand and receptor. For example, the function of CD8+T cells would be suppressed when PD-1 bind to PD-L1. Recently, increasing immune checkpoint inhibitors were invented to block the combination, thus, restore the function of immune cells.
Thank you for your suggestion.
Point 15:
-Figure 8: needs to be explained more thoroughly throughout the text. For instances which is the role of the immunosuppressive cytokines?
Response 15:
Thank you very much for your advice
-Line 306-312: As is well known that tumor immunosuppressive cytokines play a critical role in immune cell exhaustion. For instances, IL10 which secreted by M2-macrophages, could impair the proliferation and migration of T cells. Hence, we compared the expression of two immunosuppressive cytokines in high and low risk groups. Results showed that IL10 and TGFB1 were both upregulated in high-risk groups (Figure 8J), indicating that the immunosuppressive microenvironment might be further promoted through cytokines.
We appreciate a lot for you suggestion!
Discussion:
Point 16:
-Line 266: “aberrantly expressed in variable tumors”: please mention a few common tumor types.
Response 16:
Thank you very much for your careful revision
Based on your suggestion, we made a modification as following:
-Line 315: Evidence from previous research showed that collagen-related genes are aberrantly expressed in variable tumors, such as bladder cancer and prostate cancer
Thank you for your suggestion!
Point 17:
-Lines 275-276: Please explain more clearly and with details how the collagen-related genes suggested impact collagen availability and action within the tumor microenvironment and how this may have an impact in tumor progression.
Response 17:
Thanks greatly for your constructive suggestion!
Based on your suggestion, we made a modification as following:
-Line 327-331: Fibronectin 1 (FN1), an extracellular matrix glycoprotein, plays major roles in cell adhesion, migration, and differentiation [17, 18]. Remodeling of this components has been shown to contribute to invasion and metastasis, while also acting in certain tumors as a physical barrier to block immune cell infiltration, thus result in poor immune therapy response.
-Line 331-339: Interleukin-6 (IL-6) which plays a significant role in cancer progression, is a pleiotropic factor that belongs to a cytokine subfamily [20]. Evidence showed that IL-6 could mediate down-regulation of type II collagen through JAK/STAT pathway [21]. Furthermore, compared with COL VI- fibroblasts, preferential activation of IL6/STAT3 pathway were observed in COL VI+ fibroblasts, indicating IL6 contribute greatly to collagen regulation. Additionally, research showed that IL6 were highly secreted in tumor cell/fibroblast co-cultures. Blockade of IL6 in tumors co-cultured with fibroblasts resulted not only in regression of tumor growth but also in the accumulation of CD8+ TILs in intratumoral tissues.
Till now, the correlation between immune microenvironment and another three genes were seldom reported. In vivo and in vitro studies should be conducted in the future to investigate the specific mechanism between these genes and immunity.
Thank you very much for your suggestion!
Point 18:
-Explain clearly how IL-6 which is a cytokine can be considered as collagen-related gene: which is the mechanism of action?
Response 18:
Thank you for your question
In this study, we define genes involved in the biological regulation of collagen as collagen-related genes. As reported, IL6 could mediate down-regulation of type II collagen through JAK/STAT pathway. Compared with COL VI- fibroblasts, preferential activation of IL6/STAT3 pathway were observed in COL VI+ fibroblasts, indicating IL6 contribute greatly to collagen regulation. For these reasons, we embedded IL6 into this category.
Thank you very much.
We appreciate a lot for your constructive suggestion!
Point 19:
-Please whenever you mention your findings in the discussion, make a reference of the figures included in the manuscript.
Response 19:
Thank you for your suggestion!
Based on your suggestion, we made a reference of the figures included in the discussion.
Thank you very much.
Point 20:
-Lines 328-340: please explain more clearly by including more references to justify the correlation of the immunosuppression with the collagen-related genes based on your findings. Also, you should discuss more clearly the impact your findings may have in cancer immunotherapy.
Response 20:
Thank you for your suggestion!
Based on your advice, we made the modification as following:
-Line 384-393: Despite of several immune cell intrinsic mechanism of immunotherapy resistance have been described, the role of the extracellular matrix (ECM) in anti-tumor therapy remains not fully stated. Research showed that anti PD-1/PD-L1 tumors were often correlated with increased collagen and reduction of LOX12-dependent collagen depo-sition could re-sensitize tumors to PD-1/PD-L1 blockade. Besides, another report demonstrated that high collagen density could instruct macrophages to acquire an immunosuppressive phenotype (TAMs), thus, result in the low efficiency of attracting cytotoxic T cells and higher ability to inhibit T cell proliferation [32]. Moreover, TAMs were also reported to regulate tumor sustained growth by secreting collagen I, through activating of the prosurvival integrin α2β1/PI3K/AKT signaling pathway.
We appreciate your suggestion!
Point 21:
-The study of the immunosuppressive cytokines is not directly linked with your previous study in immunosuppression and the risk model. Please elaborate.
Response 21:
Thank you for your suggestion.
-Line 306-312: Immunosuppressive cytokines play a critical role in immune cell exhaustion. In malignancies, TGF-β has the capability to inhibit NK cell activity and decreasing cytokine production. Besides, IL10 is a critical immunosuppressive cytokine secreted by Treg and TAMs, which impedes the proliferation and effector T cells migration.
In this study, we found that IL10 and TGFB1 were upregulated in high-risk groups in both Training and Testing cohort, which might further promote immunosuppressive microenvironment.
Thanks greatly for your revision!
Point 22:
-Line 350-351: “Tumors have the capability to stimulate immune checkpoint targets such as LAG-3, CTLA-4, PD1, PD-L1, TIGIT and TIM-3 to protect themselves from attacking”: please explain better the mechanism of the immune checkpoints in relation to the immune/tumor cells interactions and support a link to immunotherapy.
Response 22:
Thank you for your revision!
We made the modification as following:
-Line 412-418: Except for immunosuppressive cytokines, immune checkpoints also play an important part in tumorigenesis through promotion of tumor immunosuppressive effects. For instance, Tumors have the capability to stimulate immune checkpoint targets such as PD-L1 to bind with PD-1 expressed by immune cells, thus, protect themselves from being attacked. Although immune checkpoint inhibitors (ICIs) do work, many patients are still primarily resistant to ICIs. Thus, it is necessary to figure out the mechanism of immune checkpoint inhibitor resistance. Limited research demonstrated that high ex-pression of collagen could exhaust the proportion of CD8+ T cells through activation of LAIR1 receptor, which is upregulated following CD18 interaction with collagen. Targeting impression of interaction between collagen and CD18 could enhance the sensitivity of Anti-PD-1 to lung cancer. However, the correlation between collagen and immune checkpoint has not been fully stated.
Thank your very much for your advice!
Point 23:
-Line 360-361: “could also predict the immune environment of these patients”: this is a very general term you should specify the impact of the immune microenvironment related to cancer progression and resistance to therapy. Please elaborate.
Response 23:
Thank you for your revision.
Based on your suggestion, we revised the statement as following.
-Line 428-431: Collectively, our risk model based on five collagen-related genes has a better prognostic value for ccRCC patients. Furthermore, our risk model could also predict an immunosuppressive microenvironment with high infiltration of regulatory T cells and low infiltration of M1-macrophages for ccRCC patients.
Thank you very much for your advice!
Round 2
Reviewer 2 Report
Ref: vaccines-1495533-revised
Title: A five collagen-related gene signature to estimate the prognosis and immune microenvironment in clear cell renal cell cancer
Journal: Vaccines
The manuscript entitled: “A five collagen-related gene signature to estimate the prognosis and immune microenvironment in clear cell renal cell cancer” by Shi et al. is a well written article related to the establishment of a five collagen-related gene signature to estimate the immune microenvironment and predict the prognosis of clear cell renal cell cancer patients. The article was improved after the revisions and changes made by the authors. Therefore, I believe the manuscript can now be considered for publication in the Vaccines journal.